

# Reliability and reproducibility of an Italian questionnaire on "Knowledge of high social impact Eye Diseases" (KED-IT)

Valeria Iannucci[1,*], Alice Bruscolini[1,*], Tiziano Melchiorre[2], Alessandro Lambiase[1] and Alice Mannocci[3]

[1] Department of Sense Organs, University of Roma "La Sapienza", Rome, Italy
[2] Italian Branch of the International Agency for the Prevention of Blindness - IAPB Italy ONLUS, Rome, Italy
[3] Department of Human Sciences and Quality of Life Promotion, San Raffaele University of Rome, Rome, Italy
[*] These authors contributed equally to this work.

## ABSTRACT

**Background**. Health literacy plays an important role in public health. Although this has been demonstrated in the field of ophthalmology, there are very few specific instruments available to assess eye health literacy. This work aims to develop an Italian questionnaire on knowledge of eye diseases (Knowledge on Eye Disease, Italian version; KED-IT) and to evaluate its reliability and reproducibility. The KED-IT focuses on diseases with high social impact, specifically glaucoma, macular degeneration, diabetic retinopathy and keratoconus, which is the main cause of corneal transplant in Italy.

**Methods**. A cross-sectional study was conducted. The KED-IT was self-administered by the study participants twice. The interval between each administration (T0 and T1) was 5 to 8 days. Reliability was assessed using the KR-20 coefficient. The test-retest Cohen's Kappa coefficient was estimated to measure the stability and reproducibility of the results obtained between T0 and T1.

**Results**. A total of 60 subjects participated in the study. The response rate at T1 was 92%. The KR-20 reliability coefficient of the 14-item KED-IT questionnaire was good with a value of 0.878. The Cohen's kappa value for all 14 items of the KED-IT questionnaire was $k = 0.747$, indicating good agreement.

**Conclusions**. The KED-IT is the first specific ophthalmic knowledge questionnaire validated in the Italian language and we hope that it may be a starting point for the study of eye health literacy in the Italian population.

# INTRODUCTION

It is well established that health literacy (HL) has a profound impact on public health. Health literacy as a social policy issue was postulated by Simonds in 1974. The author clearly states that "*With informed and 'health-activated' citizens, there could be a considerable reduction in morbidity and mortality*" and "*Minimum standards for 'health literacy' should*

Corresponding author
Alessandro Lambiase,
alessandro.lambiase@uniroma1.it

*be established*'' especially in the school system and through mass media communication. (*Simonds, 1974*). Over the years, HL has evolved into an increasingly complex concept. Trying to summarize the different definitions given over the years, we could define HL as the set of awareness, knowledge, and skills about health issues that enable individuals to influence their own health and quality of life in synergy with health care providers.

*Edwards et al. (2012)* postulated five stages of HL progression: a basic domain, which is inherent to knowledge; two items in the functional domain, which includes practical skills (numeracy and navigation); and two items in the higher domain, based on critical thinking (communication and decision making).

*Knowledge* involves awareness of a certain pathology and understanding of inherent issues. *Numeracy* is the ability to handle numerical information, such as cut-offs for blood tests, the normal range for systemic blood pressure, *etc. Navigation* is the ability to access care by overcoming any practical barriers, such as scheduling checkups, getting to the doctor's office, getting an interpreter if needed, *etc. Communication* is the ability to interact fluently with healthcare providers or other people about a specific health topic. Finally, *decision making* is the ability to make informed decisions about health issues (*Edwards et al., 2012*)

Individuals with low health literacy are less empowered about their health and more passive in their interactions with health care providers (*Edwards, Davies & Edwards, 2009*). Available data support that poor HL is associated with lower adherence to screening, poorer self-management of treatment (*Nagarjuna et al., 2023*), more hospitalizations (*Baker et al., 1998*), and emergency department admissions (*Griffey et al., 2014*) with an overall worsening of health outcome (*Federman et al., 2014*; *McNaughton et al., 2014*). Currently, the most widely used methods to assess HL are the Rapid Estimate of Adult Learning in Medicine (REALM) (*Davis et al., 1993*), and the Test of Functional Health Literacy in Adults (TOFHLA) (*Parker et al., 1995*) which exist in both long and short forms. Both are validated questionnaires that primarily assess reading and comprehension of written text related to general health.

HL has been widely studied in the literature, but there are very few data specific to ophthalmology (*Iskander et al., 2023*).

This study aims to assess the reliability and reproducibility of a first ophthalmic knowledge questionnaire in Italian language (Knowledge on Eye Disease, Italian version; KED-IT) focusing on four diseases with high social impact: glaucoma, macular degeneration, diabetic retinopathy and keratoconus.

## METHODS

An observational study with a cross-sectional design was used to develop and measure the feasibility and reliability of a questionnaire on knowledge of eye diseases (KED-IT).

### Participants and setting

An opportunistic sample was recruited. Subjects were contacted in the waiting rooms of the Ophthalmology Clinic and in the administrative offices of the Umberto I Hospital in Rome (Italy). The questionnaire was offered only to subjects >18 years of age who were Italian

native speakers. Subjects studying or working in ophthalmology, orthoptics or optics were excluded.

## Questionnaire development content and face validity

To ensure construct validity, each item in the questionnaire was assessed by five experts in the fields of epidemiology, community health, and ophthalmology. The experts had an academic and clinical background (Sapienza University of Rome, University San Raffaele University of Rome) or were executive officers of an experienced non-profit eye care organization operating in Italy (Italian Branch of the International Agency for the Prevention of Blindness - IAPB Italia ONLUS); the final supervision was provided by a Full Professor of Ophthalmology from Sapienza University of Rome. The experts assessed the adequacy of questions on macular degeneration, glaucoma and diabetic retinopathy based on the 2005 survey of public knowledge, attitudes, and practices related to eye health and disease KAP (*National Eye Institute & Lions Clubs International Foundation, 2007*) and proposed new questions on keratoconus.

The questionnaire was reviewed to ensure that (i) the questionnaire reflected the theoretical framework used, (ii) the items were appropriate for the construct being measured, and (iii) the items adequately covered the main study objectives. Feedback was recorded on a five-point rating checklist.

The KED-IT consisted of three sections:

- Sociodemographic: three items (gender at birth, age, education level).
- Hearsay: two items "Have you received information about eye diseases in the past year?" (y/n) and where (newspapers, television, internet, social media, parents/friends, *etc.*).
- Knowledge: 14 items on four eye diseases with high social impact. Eleven items were taken from the KAP 2005 (*National Eye Institute & Lions Clubs International Foundation, 2007*) covering three eye diseases with a significant global impact: glaucoma (*Tham et al., 2014*), macular degeneration (*Fleckenstein, Schmitz-Valckenberg & Chakravarthy, 2024*), diabetic retinopathy (*Wong & Sabanayagam, 2019*). The KAP2005 was written in English, so the items included in the KED were translated into Italian using the Brislin back-translation method by two translators who were linguistic and subject matter experts. Their translations from English were compared to obtain a common Italian version. A third researcher also translated the common version into English to assess the overlap with the original English version. KED-IT also includes three items on a fourth eye disease, keratoconus, which was not present in the KAP2005 and is the leading cause of corneal transplant in Italy (*Frigo et al., 2015*). All 14 items ask about knowledge of risk factors, possible early symptoms, risk of vision loss and possible treatments. The 14 items allowed responses in the form of "false", "true", or "not sure". A score of "1" was given for a correct response, and a score of "0" was given for an incorrect or "unsure" response. The KED-IT is shown in Table 1.

The questionnaire was pre-tested on 5 respondents to ensure consistency of wording. Some words were rephrased in layman's terms (*e.g.*, "blindness" or "visual damage" instead of "vision loss", "to keep under control" instead of "to treat") and a simple definition of

central vision was added to better explain the symptoms of macular degeneration. A direct true/false question was preferred to direct assertion to actively engage the participant and encourage a thoughtful response. A yes/no format could have been used as well, as long as both options are similar and understandable in the Italian language.

The revised items were pre-tested again with the same respondents to ensure their understanding of the revised wording. Respondents who participated in the pre-test were not included in the pilot test.

## Study administration

Researchers introduced themselves to potential participants, who were informed of the purpose of the pilot study. Written informed consent was obtained from those who agreed to participate. The questionnaire was self-administered, with assistance from the researchers to clarify items and ensure that the questionnaire was fully completed. The questionnaire was administered twice, 5 to 8 days apart (T0 and T1).

## Statistical analysis

Data were analysed using PSPP version 1.6 for Windows.

Participants' demographic background and practice information were analysed using descriptive statistics. Data were presented as means (SD) for numerical and normally distributed data, or as frequencies and percentages (%) for categorical data. The normal distribution of continuous variables in the sample was assessed using the Kolmogorov–Smirnov test.

Difficulty and discrimination indices were assessed to identify items that should be acceptable (*Mitra et al., 2009*).

The difficulty index (DI) was estimated according to the Sim & Rasiah approach (*Sim & Rasiah, 2006*). According to *Mitra et al. (2009)* the difficulty index value was considered acceptable between 30% and 70% (*Rahim, 2010*; *Mamot et al., 2021*).

The discrimination index indicates how well an instrument discriminates between high and low scorers (*Sim & Rasiah, 2006*). Its computation was performed according to *Rahim (2010)*. The interpretation of the discrimination index score can vary, in the present study was considered low discrimination ≤0.19; good discrimination ≥0.2 according to *Rahim (2010)*.

Regarding the internal consistency reliability test of the KED-IT questionnaire, the KR-20 reliability coefficient, which is a special form of Cronbach's alpha, was conducted because the responses of the questionnaire were dichotomous in nature (*Tan, 2009*). KR-20 reliability coefficients of <0.50 were interpreted as low, 0.50−0.80 as moderate, and >0.80 as high (*Tan, 2009*).

An overall KR-20 coefficient and KR-20 coefficients when an item was deleted were calculated.

For test-retest reliability, at least 26 participants (50%) should complete the questionnaire again within 5–8 days. Due to the dichotomous nature of the items responses, the robust statistical method Cohen's Kappa statistic (K) was used to estimate the reliability (*Sim & Wright, 2005*). The K result was interpreted as follows: ≤ 0 no agreement; 0.01−0.20 none

**Table 1  Questionnaire KED Italian version and English translation.**

| Section | | KED-IT (Italian language validated version) | KED (translation in English) |
|---|---|---|---|
| Socio-Demographic | | Genere alla nascita (maschio/femmina) | Gender at birth (male/female) |
| | | Età in anni compiuti | Age (years) |
| | | Livello educativo (elementari o medie/superiori/ laurea o più) | Education (Elementary school; middle school; high school; Bachelor or upper) |
| Hearsay | | Nell'ultimo anno hai sentito parlare di malattie degli occhi? (sì/no) | In the past year, have you received information about eye diseases? |
| | | Se sì, attraverso quale mezzo di informazione? (indicare il principale) - riviste/giornali - opuscoli educativi - televisione - radio - internet/social media - organizzazioni sociali/religiose - In uno studio medico/clinica/durante uno screening sanitario - Sul posto di lavoro - da parenti/amici | If so, from what source of information? (indicate the main one) - Magazines/newspapers - Educational pamphlets - Television - Radio - Internet/social media - Social or religious organizations - At a doctor's office, clinic, or community health screening - At your workplace - From relatives or friends |
| | *Introduction* | *Ti faremo alcune domande su quattro importanti malattie degli occhi. Ti chiediamo di rispondere a ciascuna di esse:* | *We will now ask you some questions about four important eye diseases. Please answer each one:* |
| | | *Adesso parliamo di una malattia chiamata glaucoma:* 1) Il glaucoma può causare cecità? (vero* / falso/non so) | Now let's talk about a condition called glaucoma: 1) Can glaucoma cause blindness? (true* /false/ I don't know) |
| | | 2) Il glaucoma ha dei sintomi iniziali di cui mi posso accorgere? (vero/ falso* /non so) | 2) Does glaucoma have any early symptoms that I can notice? (true/false* / I don't know) |
| | Glaucoma | 3) I danni alla vista causati dal glaucoma possono essere prevenuti? (vero* / falso/non so) | 3) Can visual damage caused by glaucoma be prevented? (true* /false/ I don't know) |
| | | 4) Esistono terapie per tenere sotto controllo il glaucoma? (vero* / falso/non so) | 4) Are there therapies available to keep glaucoma under control? (true* /false/ I don't know) |
| | | *Adesso parliamo di una malattia chiamata maculopatia:* 5) Le vitamine e lo zinco possono essere d'aiuto nella maculopatia? (vero* / falso/non so) | *Now let's talk about a condition called macular degeneration:* 5) Can vitamins and zinc help with macular degeneration? (true* /false/ I don't know) |
| | | 6) La maculopatia può colpire più membri della stessa famiglia? (vero* / falso/non so) | 6) Can macular degeneration run in families? (true* /false/ I don't know) |

**Table 1** (*continued*)

| Section | | KED-IT (Italian language validated version) | KED (translation in English) |
|---|---|---|---|
| | Macular degeneration | 7) Una persona può essere affetta da maculopatia senza saperlo? (vero*/ falso/non so) | 7) Can a person have macular degeneration and not know it? (true*/false/ I don't know) |
| | | 8) La vista centrale è quella che ti permette di leggere, riconoscere i visi delle persone e vedere il cibo nel piatto. Sapendo questo, secondo te la maculopatia può causare la perdita della vista centrale? (vero*/ falso/non so) | 8) Central vision is what allows you to read, recognize people's faces, and see the food on your plate. Knowing this, do you think macular degeneration can affect central vision? (true*/false/ I don't know) |
| Knowledge | Diabetic retinopathy | *Adesso parliamo di una malattia chiamata retinopatia diabetica:* 9) Le persone con diabete sono più a rischio di malattie degli occhi rispetto alle persone senza diabete? (vero*/ falso/non so) | *Now let's talk about a condition called diabetic retinopathy:* 9) Do people with diabetes have a higher risk of eye disease than people without diabetes? (true*/false/ I don't know) |
| | | 10) è vero che le persone con diabete dovrebbero sottoporsi a una visita oculistica con le gocce per dilatare la pupilla almeno una volta l'anno? (vero*/ falso/non so) | 10) Is it true that people with diabetes should have an eye exam with pupil dilating drops at least once a year? (true*/false/ I don't know) |
| | | 11) è vero che NON esistono farmaci o altre terapie per tenere sotto controllo le malattie degli occhi causate dal diabete? (vero/ falso*/non so) | 11) Is it true that there are NO medications or other therapies to control eye disease caused by diabetes? (true/false */ I don't know) |
| | Keratoconus | *Adesso parliamo di una malattia chiamata cheratocono:* 12) Il cheratocono può danneggiare gravemente la vista? (vero*/ falso/non so) | *Now let's talk about a condition called keratoconus:* 12) Can keratoconus seriously affect vision? (true*/false/ I don't know) |
| | | 13) Il cheratocono ha dei sintomi iniziali di cui mi posso accorgere? (vero*/ falso/non so) | 13) Does keratoconus have any early symptoms that I can notice? (true*/false/ I don't know) |
| | | 14) Esistono terapie per curare il cheratocono? (vero*/ falso/non so) | 14) Are there any therapies for keratoconus? (true*/false/ I don't know) |

**Notes.**

*The correct answer (not shown to participants).

to slight; 0.21−0.40 fair; 0.41−0.60 moderate agreement; 0.61−0.80 substantial agreement; 0.81−1.00 near perfect agreement (*Sim & Wright, 2005*). A K value for each item and an average kappa value were calculated.

A $p$-value <0.05 was considered statistically significant.

## Sample Size

The sample size for the internal consistency reliability analysis using the KR-20 was calculated based on the subject to item ratio, using a subject to item ratio of 4:1 (*Tan, 2009*). Therefore, a minimum of 56 participants was required (14 items × 4 = 56).

In terms of test-retest reliability, the sample size for testing Cohen's kappa agreement was determined to be 26, which is 50% of the total number of participants (*Bujang & Baharum, 2022*).

### Ethics

The study was conducted in conformity with the Declaration of Helsinki (*The World Medical Association, 2008*).

The protocol of this study was reviewed and approved by the Institutional Review Board of Sapienza University of Rome, Department of Psychology (Approval number: 742).

## RESULTS

### Respondent characteristics

68 individuals were invited to participate; 60 of them agreed to participate (88.2%). The mean age of the 8 individuals who did not participate in the study was 41 years (SD =11), and 5 (62.5%) were male. The age distribution of the respondents was Normal (the Kolmogorov Smirnov test's significant level was $p = 0.459$), the mean age was 41.6 (SD = 15.6) years, and more than half (58.3%) were female. 45% of the respondents had completed higher education (university degree) and 55% had completed high school.

The mean time to complete the KED-IT, as assessed by the questionnaire administrator, was 7 min ± 2 min.

### Reliability, difficulty and discrimination of KED-IT

Table 2 shows the distribution of correct answers, the Difficulty Index (DI) and Discrimination Index for each KED-IT item.

In terms of DI, 1 item was high difficulty, ("Does glaucoma have any early symptoms that I can notice?"); 7 items were acceptable ("Can vitamins and zinc help with macular degeneration?", "Can macular degeneration run in families?", "Can a person have macular degeneration and not know it?", "Central vision is what allows you to read, recognize people's faces, and see the food on your plate. Knowing this, do you think macular degeneration can affect central vision?", "Can keratoconus seriously affect vision?", "Does keratoconus have any early symptoms that I can notice?" and "Are there therapies for keratoconus?") ; 6 items were easy ("Can glaucoma cause blindness?", "Can visual damage caused by glaucoma be prevented?", "Are there therapies available to keep glaucoma under control?", "Do people with diabetes have a higher risk of eye disease than people without diabetes?", "Is it true that people with diabetes should have an eye exam with pupil dilating drops at least once a year?" and "Is it true that there are NO medications or other therapies to control eye disease caused by diabetes?").

A good discrimination index (>0.19) was reported by 11 out of 14 items. Item 2 ("Does glaucoma have early symptoms that I can notice?"), item 3 ("Can visual impairment caused by glaucoma be prevented?") and item 4 ("Are there therapies available to keep glaucoma under control?") reported a high discrimination index of 0.08, 0.10 and 0.13, respectively.

The KR-20 internal consistency reliability coefficient of the 14-item KED-IT questionnaire was 0.878 (Table 2). The KR-20 alpha coefficient was good. All items

**Table 2  Item analysis for eye disease knowledge: description of the difficulty index (DI) and discrimination index (N=60).**

| KED-IT Disease Items | Correct answer (N) | DI (%) | DI level[a] | Discrimination Index [c d] | KR-20[b] |
|---|---|---|---|---|---|
| (1) Can glaucoma cause blindness? | 51 | 85.0 | 1 | 0.23 | 0.865 |
| (2) Does glaucoma have any early symptoms that I can notice? | 13 | 21.7 | 3 | 0.08[d] | 0.889 |
| (3) Can visual impairment caused by glaucoma be prevented? | 50 | 83.3 | 1 | 0.10[d] | 0.870 |
| (4) Are there therapies available to keep glaucoma under control? | 55 | 91.7 | 1 | 0.13[d] | 0.872 |
| (5) Can vitamins and zinc help with macular degeneration? | 24 | 40.0 | 2 | 0.42 | 0.879 |
| (6) Can macular degeneration run in families? | 33 | 55.0 | 2 | 0.43 | 0.865 |
| (7) Can a person have macular degeneration and not know it? | 35 | 58.3 | 2 | 0.33 | 0.875 |
| (8) Central vision is what allows you to read, recognize people's faces, and see the food on your plate. Knowing this, do you think macular degeneration can affect central vision? | 41 | 68.3 | 2 | 0.28 | 0.873 |
| (9) Do people with diabetes have a higher risk of eye disease than people without diabetes? | 44 | 73.3 | 1 | 0.33 | 0.863 |
| (10) Is it true that people with diabetes should have an eye exam with pupil dilating drops at least once a year? | 46 | 76.7 | 1 | 0.28 | 0.864 |
| (11) Is it true that there are NO medications or other therapies to control eye disease caused by diabetes? | 43 | 71.7 | 1 | 0.38 | 0.863 |
| (12) Can keratoconus seriously affect vision? | 23 | 38.3 | 2 | 0.43 | 0.867 |
| (13) Does keratoconus have any early symptoms that I can notice? | 23 | 38.3 | 2 | 0.40 | 0.867 |
| (14) Are there therapies for keratoconus? | 30 | 50.0 | 2 | 0.48 | 0.869 |
| Overall KR-20 | | | | | **0.878** |

**Notes.**
[a] 1= item considered easy (>70%); 2= item considered acceptable (30–70%); 3=item considered difficult (<30%)
[b] If item was deleted.
[c] $N_U$=20 $N_L$= 20.
[d] Low discrimination $\leq 0.19$; good discrimination >0.19.

showed the same contribution to the overall reliability: when one item was removed, the KR-20 was essentially the same: it ranged from 0.863 to 0.889.

## Reproducibility of KED-IT

The 60 subjects who answered the questionnaire (T0) were asked to answer it again 5 to 8 days later (T1). The test-retest was fulfilled for 55 subjects with a response rate of 92%. 5 subjects were no longer available, and it was not possible to administer the T1 questionnaire: they were four females, and the mean age was 61 years (SD =5.1 years).

The results of the test-retest reliability of the KED-IT questionnaire are shown in Table 3.

Correlation coefficients between the two administrations (T0 *versus* T1) ranged from 0.50 to 0.99, with 8 out of 14 items showing more than 0.7, indicating very good reproducibility (Table 3).

**Table 3** Test-retest reliability of the KED-IT questionnaire between T0 and T1 questionnaire administration.

| Item | Kappa value[a] | p |
|---|---|---|
| 1) Can glaucoma cause blindness? | 0.791 | <0.001 |
| 2) Does glaucoma have any early symptoms that I can notice? | 0.657 | <0.001 |
| 3) Can visual impairment caused by glaucoma be prevented? | 0.513 | <0.001 |
| 4) Are there therapies available to keep glaucoma under control? | 0.627 | <0.001 |
| 5) Can vitamins and zinc help with macular degeneration? | 0.679 | <0.001 |
| 6) Can macular degeneration run in families? | 0.742 | <0.001 |
| 7) Can a person have macular degeneration and not know it? | 0.628 | <0.001 |
| 8) Central vision is what allows you to read, recognize people's faces, and see the food on your plate. Knowing this, do you think macular degeneration can affect central vision? | 0.539 | <0.001 |
| 9) Do people with diabetes have a higher risk of eye disease than people without diabetes? | 0.908 | <0.001 |
| 10) Is it true that people with diabetes should have an eye exam with pupil dilating drops at least once a year? | 0.999 | <0.001 |
| 11) Is it true that there are NO medications or other therapies to control eye disease caused by diabetes? | 0.866 | <0.001 |
| 12) Can keratoconus seriously affect vision? | 0.769 | <0.001 |
| 13) Does keratoconus have any early symptoms that I can notice? | 0.888 | <0.001 |
| 14) Are there any therapies for keratoconus? | 0.854 | <0.001 |

**Notes.**
[a]$k = 0.4 - 0.60$ moderate agreement; $k = 0.61 - 0.8$ substantial agreement; $k > 0.8$ excellent agreement

The average kappa value for all 14 items of the KED-IT questionnaire was $k = 0.747$, of which 3 items showed moderate agreement (k: 0.4−0.6), 6 good (k: 0.61−0.8) and 5 excellent agreement ($k > 0.8$).

## DISCUSSION

The burden of eye disease on public health and quality of life is far from negligible.

Glaucoma is the leading cause of irreversible blindness worldwide, affecting 3.5% of the population over the age of 40 (*Tham et al., 2014*). Age-related macular degeneration (AMD) affects 196 million people worldwide and is becoming increasingly prevalent due to the general aging of the population (*Fleckenstein, Schmitz-Valckenberg & Chakravarthy, 2024*). Diabetic retinopathy is the leading cause of vision loss and blindness in the working-age population; specifically, diabetic macular edema (DME) affects approximately 6.8% of adults with diabetes between the ages of 20 and 79 (*Wong & Sabanayagam, 2019*). Unfortunately, epidemiologic data assessing glaucoma and macular degeneration in Italy are lacking. Keratoconus, although considered a rare disease, is the first cause of corneal transplant in Italy (*Frigo et al., 2015*). Specifically, 42.5% of penetrating keratoplasty (PK)

and 69.6% of anterior lamellar keratoplasty (ALK) are performed for keratoconus in Italy (*Frigo et al., 2015*).

Despite the high social impact of eye diseases, there are no specific HL validated questionnaires in Italian.

The KED-IT is a potentially useful eye disease knowledge questionnaire for a general population: it has good overall reliability (KR-20 >0.80) and good retest reproducibility (Cohen's K ranged from moderate to perfect) with an average kappa value for all 14 items of 0.747.

Regarding the difficulty and discriminant indices, 13 items were acceptable with at least one index level in the Rahim range (*Rahim, 2010*) and can be retained in a questionnaire. Only the item 2 ("Does glaucoma have any early symptoms that I can notice?") contributed to a modest reduction in KR, as shown in Table 2. Nevertheless, considering that the reduction in KR was small (0.011), item 2 is highly relevant for assessing glaucoma awareness, and glaucoma has a significant impact on society (*Tham et al., 2014*) all authors agreed to retain it in the KED-IT.

Regarding DI and Discrimination Index, items 2, 3, and 4 about glaucoma had low discriminatory values (<0.19) and different proprotion of difficulties as shown in Table 2. This highlights that some aspects of glaucoma are extremely well known (items 3 and 4, DI = 83.3% and 91.7%, respectively), while others are almost unknown (item 2, DI =21.7%) in our study population. Since all three of these questions ("Does glaucoma have any early symptoms that I can notice?", "Can visual damage caused by glaucoma be prevented?", "Are there therapies available to keep glaucoma under control?") are important for assessing knowledge about such an impactful disease, it was deemed appropriate to include them in the questionnaire.

The KED-IT showed the advantage of being easy to understand and quick to administer: none of the participants complained of fatigue or doubts while completing the questionnaire.

Nevertheless, this pilot study has some limitations. Our study population is highly educated and recruited only in hospital waiting rooms and administrative offices, so may not be representative of the Italian general population.

To make the questions easier to understand, it was necessary to simplify and generalize some complex topics; for example, the general term "maculopatia" (macular degeneration) was chosen regardless of the specific type of macular degeneration, which also varies widely in prevalence, prognosis, and treatment. This approximation was accepted because the term "maculopatia" in common Italian language refers principally to age-related macular degeneration (AMD).

Although there is room for improvement, the KED-IT is the first questionnaire in Italy (and one of the few in the world) that has been specifically validated to assess knowledge of the most impactful eye diseases.

The most difficult item ("Does glaucoma have early symptoms that I can notice?") may be the one that explains why glaucoma is such a threat to public health. This preliminary analysis shows that 78.3% of respondents are unaware that glaucoma is a silent disease in its early stages. This may lead to late diagnosis with severe deterioration in visual outcomes.

If confirmed in larger samples representative of the general population, this finding will highlight the need for more widespread and effective community education to prevent glaucoma blindness.

## CONCLUSIONS AND NEXT STEPS

Health literacy has a great impact on public health but, despite the high burden of eye diseases, there is very little specific work on eye health literacy.

The Knowledge on Eye Disease, Italian version (KED-IT) is the first validated questionnaire to assess knowledge of impactful eye diseases in Italy (glaucoma, macular degeneration, diabetic retinopathy and keratoconus).

The KED-IT has good reliability and reproducibility, proving to be a valid questionnaire to assess knowledge of eye diseases in our study population.

If further tested and refined on a larger sample representative of the Italian population, the KED-IT may prove to be a useful tool for assessing knowledge of major eye diseases, targeting educational resources to the most underserved Italian communities and overall improving eye health in Italy.

### Funding

This study was funded by the European Union–FSE REACT.EU, PON Research and Innovation 2014-2020. There was no additional external funding received for this study. The funders had no role in study design, data collection and analysis, decision to publish, or preparation of the manuscript.

### Grant Disclosures

The following grant information was disclosed by the authors:
the European Union–FSE REACT.EU, PON Research and Innovation 2014-2020.

### Competing Interests

The authors declare there are no competing interests.

### Author Contributions

- Valeria Iannucci conceived and designed the experiments, performed the experiments, prepared figures and/or tables, and approved the final draft.
- Alice Bruscolini performed the experiments, authored or reviewed drafts of the article, and approved the final draft.
- Tiziano Melchiorre analyzed the data, authored or reviewed drafts of the article, and approved the final draft.
- Alessandro Lambiase conceived and designed the experiments, authored or reviewed drafts of the article, and approved the final draft.
- Alice Mannocci conceived and designed the experiments, analyzed the data, prepared figures and/or tables, and approved the final draft.

## Human Ethics

The following information was supplied relating to ethical approvals (*i.e.*, approving body and any reference numbers):

The Institutional Review Board, Department of Psychology, Sapienza University of Rome approved the study (742).

## Data Availability

Raw data are available in the Supplemental Files.

## Supplemental Information

Supplemental information for this article can be found online at http://dx.doi.org/10.7717/peerj.17906#supplemental-information.

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
