# Peer review of "Reliability and reproducibility of an Italian questionnaire on "Knowledge of high social impact Eye Diseases" (KED-IT)"

_PeerJ, doi:10.7717/peerj.17906_

## Round 0.1 · original submission · Minor Revisions

Three reviewers have assessed the manuscript and for the most part the reviews are very positive. Please address all of the reviewer comments carefully. In particular, please more carefully define the qualifications of those individuals who validated the survey. Also, please consider what material can be moved from the introduction to the discussion. Of most significance, please summarize the "next steps" in improving and implementing the questionnaire. Specifically, what things were learned in this study that help in moving things forward.

Reviewer 1 ·

Basic reporting

In the introduction first 2 paragraphs, the author elaborated the findings of Edwards (2012) and Simons (1974), which my be improved to become more scientific.

It is advisable for the author to include the prevalence of the selected eye diseases in Italy to further justify the selection of eye diseases to be included in the questionnaire. The justification for keratoconus is an exemplary.

Experimental design

"The KAP2005 was written in English, so the items included in the KED were translated into Italian using the Brislin back-translation method by two translators who were linguistic and subject matter experts. Their translations were compared to obtain a common version.
Question " their translation were compared between each other?"

A third researcher also translated the common version into English to assess the overlap with the original
version.
Question " which common version is this sentence referring to?" or was the newly Italian version translated back to English, to assess the overlap? " . The sentence is not so clear.

Validity of the findings

This item also improved the KR 20 coefficient when it was deleted (KR-20 =0.889). For these reasons, the authors considered the possibility of removing item 2 from the questionnaire. However, because this question is highly relevant in assessing glaucoma awareness, it was decided to keep item 2 in the KED-IT.
Suggestion " The consideration to remove item 2 is highly recommendable".

Additional comments

This is a good pilot study to embark on further investigations on the awareness on eye diseases in other populations in the country.

Reviewer 2 ·

Basic reporting

The paper "Reliability and reproducibility of an Italian questionnaire on Knowledge of high social impact Eye Diseases” (KED-IT)," by Valeria Iannucci et al. is a well conceived and executed study aimed at developing an instrument to assess Eye Health Literacy in Italian. This questionnaire, "Knowledge of Eye Diseases" (KED-IT), could be a useful patient care, education, and research tool.

Experimental design

The design appears to be appropriate. An opportunistic sample is appropriate for a pilot survey study. However, the number of subjects that accepted vs declined should ideally be reported. Perhaps more detail on the types of patients found in these clinical setting could be provided.

Validity of the findings

The methods and analysis are well done.

Methods.
“... ensure construct validity, each construct in the questionnaire was validated by experts in the fields of epidemiology, community health, and ophthalmology." Please be more specific as to the qualifications of these individuals. What made them experts? Were they at universities settings? Are they board certified in these different fields? Did they validate the questions or simply review them to ensure that they were appropriate?

Was age normally distributed. If not, SD is not the most appropriate descriptor.

How was completion time for survey assessed? Was it by means of an independent monitor? This detail is omitted.

Additional comments

The paper is well written but introduction too long. It needs to lay out the problem, or efficiently put into context what gap the present works serves and get onto it. Move the excess material to the discussion, as appropriate. Justifying the choice of eye diseases is better left to the discussion. This would make the paper more efficient and readable. Placing information in the introduction about the scope of individual diseases is unnecessary and too specific.

In the introduction the definition of HL from Simonds in 1974 is referenced, but not stated (Simonds, 1974) I do not have access to this source. What is the historic definition?

According to Iskander and colleagues, HL has been widely studied in the literature, but there are very few data specific to ophthalmology (Iskander et al., 2023). —> style why reference a specific senior author by name? This point should just be made more generally.

It would strengthen the paper and I believe it reasonable to rephrase the first sentence of the introduction: “The KED-IT IS a potentially useful eye disease knowledge questionnaire for -A- general population”

Discussion

Our "sample" => replace with study "population"

Conclusions and next steps is lacking.

This should be added to the paper.

They "key points" for the article would make a solid basis for the conclusion and future directions

Reviewer 3 ·

Basic reporting

Lines 60-61: Is the statement cited from Frigo et al., 2015 as well? It requires citation.
"Keratoconus, although considered a rare disease, is the first cause of corneal transplant in Italy."

Overall, the article is well written, setting a clear background why this study is conducted.

Experimental design

The aims and the questionnaires conform to the study's locality. The inclusion of keratoconus in the questionnaire is well justified. It is always important that the study can highlight what is important and beneficial for the surrounding community.

From the reader's understanding, the first 11 questions were adapted from the KAP 2005 questionnaire of the National Eye Institute (lines 91-94). It would be helpful for readers if the authors considered introducing a clearer setting for this developed KED-IT.
Suggestion: The knowledge section has 14 items on four eye diseases with high social impact. The first 11 items on glaucoma, macular degeneration, and diabetic retinopathy were adopted from the KAP 2005 and translated from English to Italian. Three items on keratoconus disease were added to KED-IT, considering that it is the leading cause of corneal transplants in Italy.

Table 1:
Although the questions were already published in the KAP 2005, the reader thinks the question is better for a yes or no answer. E.g., Can glaucoma cause blindness? (Yes/ No/ I don't know). The answer true/ false is more suitable with a statement style, i.e., Glaucoma causes blindness (True/ False/ I don't know)

As this study has been completed, consider addressing this part as a limitation or things to be improved before implementing the KED-IT for the Italian population.

Is the questionnaire given to the participant have the * indicating the correct answer?

Lines 167-168: Please put in the complete name of the ethical review board. e.g., Institutional Review Board of University XX (approval no: 742).

Validity of the findings

The findings are helpful and important for improving future studies. The reader supported the idea of using a common Italian language that is easier to understand by the Italian population (e.g., maculopatia).

Additional comments

This study highlights the local findings, which are very important in public health for a focused intervention in eye health awareness.

---

## Round 0.2 · accepted · Accept

Thank you for addressing the reviewer comments. I found the material in the introduction informative, so I am recommending that the introduction remain as it is.

Reviewer 1 ·

Basic reporting

My comments were well addressed. May further investigations be carried out in relation to those diseases in Italy in the near future.

Experimental design

No comment

Validity of the findings

No comment

Reviewer 2 ·

Basic reporting

There is clear, unambiguous, and professional English used throughout. However, the article could be shortened.

Experimental design

Sufficiently updated. The revisions are adequate

Validity of the findings

The Data is clearly presented

Additional comments

The article could be improved by being made more concise. I am not certain that the background material added (as requested by another reviewer) is needed, especially given that it does not relate to Italy.

Reviewer 3 ·

Basic reporting

No comment

Experimental design

Acceptable. Justification and improvement on the methods raised previously have been made.

Validity of the findings

The findings will add to the body of knowledge addressing the local needs of the study.

Additional comments

No further comment.